# Amortized Bayesian Decision Making
# for simulation-based models

**Mila Gorecki**                                           *mila.gorecki@tuebingen.mpg.de*
*Social Foundations of Computation, Max Planck Institute for Intelligent Systems, Tübingen*
*Tübingen AI Center*

**Jakob H. Macke**                                          *jakob.macke@uni-tuebingen.de*
*Machine Learning in Science, Excellence Cluster Machine Learning, University of Tübingen*
*Empirical Inference, Max Planck Institute for Intelligent Systems, Tübingen*
*Tübingen AI Center*

**Michael Deistler**                                      *michael.deistler@uni-tuebingen.de*
*Machine Learning in Science, Excellence Cluster Machine Learning, University of Tübingen*
*Tübingen AI Center*

**Reviewed on OpenReview:** *https://openreview.net/forum?id=BQE4MTAfCE*

## Abstract

Simulation-based inference (SBI) provides a powerful framework for inferring posterior distributions of stochastic simulators in a wide range of domains. In many settings, however, the posterior distribution is not the end goal itself — rather, the derived parameter values and their uncertainties are used as a basis for deciding what actions to take. Unfortunately, because posterior distributions provided by SBI are (potentially crude) approximations of the true posterior, the resulting decisions can be suboptimal. Here, we address the question of how to perform Bayesian decision making on stochastic simulators, and how one can circumvent the need to compute an explicit approximation to the posterior[1]. Our method trains a neural network on simulated data and can predict the expected cost given any data and action, and can, thus, be directly used to infer the action with lowest cost. We apply our method to several benchmark problems and demonstrate that it induces similar cost as the true posterior distribution. We then apply the method to infer optimal actions in a real-world simulator in the medical neurosciences, the Bayesian Virtual Epileptic Patient, and demonstrate that it allows to infer actions associated with low cost after few simulations.

## 1 Introduction

Many quantities in the sciences and engineering cannot be directly observed, but knowledge of these quantities (often called 'parameters' $\boldsymbol{\theta}$) is crucial to make well-informed *decisions* on how to design experiments, claim discoveries, or assign medical treatments. Take, for example, a mechanistic model in a medical context that describes a specific brain function, but whose parameters are not directly or easily measurable. Given observations from patients, various methods have been proposed to determine likely parameter values and their uncertainty. Ultimately, however, the goal in this scenario is to assign medical diagnoses or to decide whether an intervention is needed. Therefore, we have to turn our beliefs about parameter values into concrete and accurate actions. To do so, Bayesian decision theory provides a framework to select optimal decisions (or 'actions') under noisy and potentially non-linear measurements of the parameters (Berger, 2013). Given the measurement $\mathbf{x}_o$ and a cost function that assigns a *cost* to each action (for example corresponding to the

---

[1]Concurrently to our work, another work on Bayesian decision making for simulators has been published (Alsing et al., 2023). See Discussion for similarities and differences of our works.

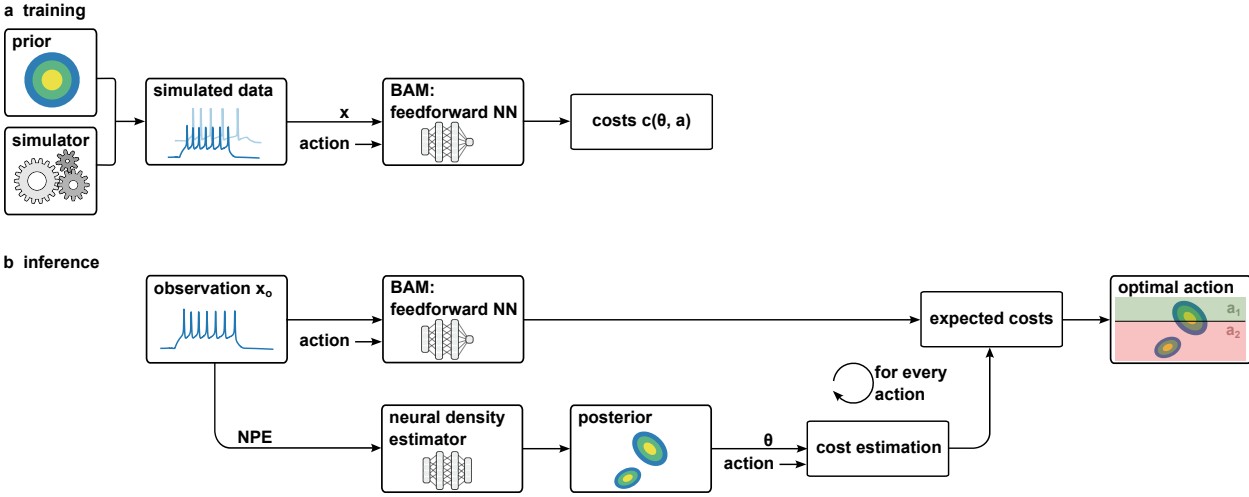

Figure 1: **Amortized Bayesian Decision Making.** Given a (stochastic) simulator and a prior over parameters, we propose to learn a neural network which directly estimates the expected cost associated with any possible observation and action, i.e. a 'Bayesian Amortized Decision Making' (BAM) network. (a) Analogous to simulation-based methods like Neural Posterior Estimation (NPE), BAM starts by generating a database of (parameter, data) pairs by drawing prior samples and simulating them. BAM then uses a feedforward neural network trained to directly predict the estimated costs of taking a particular action given observations. This way, BAM circumvents the computation of an explicit posterior approximation. (b) At inference, BAM requires only a single pass through the network to estimate the expected posterior costs of each data-action-pair $(x_o, a)$. In contrast, NPE builds an explicit parameter estimate of the posterior. This can be used to estimate the expected cost of a given action through Monte-Carlo estimation (NPE-MC). For both algorithms, we select the action which leads to the lowest expected costs.

associated risk of each intervention), Bayesian decision making chooses the action which induces the lowest cost, averaged over the distribution of parameters given the measurements $p(\boldsymbol{\theta}|\mathbf{x}_o)$:

$$\mathbf{a}^* = \arg\min_{\mathbf{a}} \int c(\boldsymbol{\theta}, \mathbf{a}) \cdot p(\boldsymbol{\theta}|\mathbf{x}_o) d\boldsymbol{\theta}.$$

The main difficulty associated with Bayesian decision making is to infer the posterior distribution $p(\boldsymbol{\theta}|\mathbf{x}_o)$ over (potentially many) parameters. The posterior distribution

$$p(\boldsymbol{\theta}|\mathbf{x}_o) \propto p(\mathbf{x}_o|\boldsymbol{\theta})p(\boldsymbol{\theta})$$

is intractable for all but the simplest models, and requires methods such as Markov-Chain Monte Carlo (MCMC) or variational inference to approximate. However, these methods require that the likelihood can be evaluated, but many models in the sciences can only be simulated (Gonçalves et al., 2020; Cranmer et al., 2020), i.e., their likelihood can only be sampled from. In addition, such methods require to re-run inference for new data, which is computationally expensive. Contrary to that, recent methods from the field of 'simulation-based inference' (SBI) (Cranmer et al., 2020) require a large upfront cost to generate a database of simulations (but no likelihood evaluations), and they can then infer the posterior distribution given any data without further simulator runs (Hermans et al., 2020; Gonçalves et al., 2020; Radev et al., 2020).

Here, we first investigate how faithful the actions induced by a particular amortized SBI algorithm, 'Neural Posterior Estimation' (NPE) (Papamakarios and Murray, 2016; Greenberg et al., 2019), are. We demonstrate that, in cases where the posterior is inaccurate (Lueckmann et al., 2021; Hermans et al., 2022), the induced 'Bayes-optimal' action will also be (largely) inaccurate, and that this failure can occur even for simulators with few free parameters (Lacoste–Julien et al., 2011). In addition, while NPE amortizes simulation and

training, it still has to compute the (potentially expensive) cost of several (parameter, action) pairs for every observed data, which incur a substantial additional cost at decision time.

To address these limitations, we propose a method which we term **B**ayesian **A**mortized decision **M**aking (BAM) (Fig. 1). Like NPE, BAM uses an (upfront generated) database of simulations to train a neural network. Unlike NPE, however, BAM circumvents the need to learn the (potentially high-dimensional) posterior distribution explicitly and instead only requires to train a feedforward neural network which is trained to directly predict the expected costs for any data and action. At inference time, BAM requires only one forward pass through the neural network, without the requirement of computing the cost by averaging over the posterior. Because of this, BAM can perform amortized Bayesian decision making in milliseconds even in cases where the cost function itself is expensive to compute.

We define decisions on four benchmark tasks (Lueckmann et al., 2021) and evaluate both algorithms on all of those tasks. We demonstrate that, on tasks where NPE is known to perform poorly (and returns a poor estimate of the posterior), our alternative method, BAM, can lead to significantly better decisions, sometimes leading to a reduction of cost by almost an order of magnitude. We then apply both methods to a real-world problem from medical neuroscience: given potentially epileptic measured brain activity, we infer which treatment option would be most suitable. We demonstrate that, given a specified loss function, both algorithms can identify optimal decisions with only a few simulations.

## 2 Background

### 2.1 Problem setting

We consider a stochastic simulator with free parameters $\boldsymbol{\theta}$ and assume access to a prior distribution $p(\boldsymbol{\theta})$. Running the (potentially computationally expensive) simulator allows generating samples corresponding to the likelihood $\mathbf{x} \sim p(\mathbf{x}|\boldsymbol{\theta})$. We assume that the likelihood is only defined implicitly through the simulator and cannot be evaluated. Given observed data $\mathbf{x}_o$ we aim to obtain the Bayes-optimal action

$$\mathbf{a}^* = \arg\min_{\mathbf{a}} \int c(\boldsymbol{\theta}, \mathbf{a}) \cdot p(\boldsymbol{\theta}|\mathbf{x}_o) d\boldsymbol{\theta},$$

where we used the posterior $p(\boldsymbol{\theta}|\mathbf{x}_o)$, defined as

$$p(\boldsymbol{\theta}|\mathbf{x}_o) \propto p(\mathbf{x}_o|\boldsymbol{\theta})p(\boldsymbol{\theta}),$$

and we defined a *cost function* $c(\boldsymbol{\theta}, \mathbf{a})$ (Schervish, 2012). This cost function quantifies the consequences of taking a particular action $\mathbf{a}$ if the true parameters of the system $\boldsymbol{\theta}$ were known. The cost function can be an (easy to compute) analytical expression of parameters and actions, but it can also require (computationally expensive) computer simulations (Alsing et al., 2023). The Bayes-optimal action is found by minimizing over actions $\mathbf{a}$, sampled from a distribution $p(\mathbf{a})$. We choose $p(\mathbf{a})$ such that its support covers all possible actions, i.e. it also contains the true optimal action. This is reasonable, as the range of available measures is usually known to the decision maker.

To evaluate the performance of different decision making algorithms, we use the 'incurred cost' of an action. Given a synthetically generated observation $\mathbf{x}_o$ and its ground truth parameters $\boldsymbol{\theta}^{\text{gt}}$, the incurred cost of an action $\mathbf{a}$ is defined as

$$c_i = c(\boldsymbol{\theta}^{\text{gt}}, \mathbf{a}).$$

### 2.2 Neural Posterior Estimation

The posterior distribution $p(\boldsymbol{\theta}|\mathbf{x}_o)$ is intractable for all but the simplest models. Neural Posterior Estimation (NPE) is a method that overcomes this by using samples from the joint distribution $p(\boldsymbol{\theta}, \mathbf{x})$, obtained by sampling $\boldsymbol{\theta}$ from the prior and simulating to learn a parametric density for the posterior distribution (Papamakarios and Murray, 2016). Concretely, NPE trains a deep neural density estimator by minimizing its negative log-likelihood:

$$\mathcal{L}(\boldsymbol{\phi}) = -\frac{1}{N} \sum_{i=1}^{N} \log q_{\boldsymbol{\phi}}(\boldsymbol{\theta}_i|\mathbf{x}_i),$$

where $\phi$ are learnable parameters. After training, the deep neural density estimator can be evaluated for any observation $\mathbf{x}_o$ and will return an estimate for the posterior distribution, which can be sampled and whose log-probability can be computed for any $\boldsymbol{\theta}$. Crucially, once the deep neural density estimator has been trained, it can be applied to *any* observation without running further simulations and without retraining, thereby *amortizing* the cost of inference.

## 3  Methods

We propose two methods to obtain *amortized* Bayesian decisions (Fig. 1). We discuss advantages and disadvantages of both methods in Sec. 5.

### 3.1  Neural Posterior Estimation and Monte-Carlo Sampling (NPE-MC)

The first method is a direct extension of Neural Posterior Estimation (NPE) and serves as a baseline. We use two properties of NPE, namely that it is (1) amortized across observations and (2) returns a posterior estimate that can be sampled quickly (1000s of samples in milliseconds in typical applications). The method proceeds in three steps: (1) train NPE to obtain an amortized neural density estimator $q_{\boldsymbol{\phi}}(\boldsymbol{\theta}|\mathbf{x})$ and (2) draw $M$ samples from the approximate density $q_{\boldsymbol{\phi}}(\boldsymbol{\theta}|\mathbf{x}_o)$ and (3) compute a Monte-Carlo estimate for the expected cost for actions $\mathbf{a}$:

$$\mathbb{E}_{p(\boldsymbol{\theta}|\mathbf{x}_o)}[c(\boldsymbol{\theta},\mathbf{a})] \approx \frac{1}{M}\sum_{i=1}^{M} c(\boldsymbol{\theta}_i,\mathbf{a})$$

If the action $\mathbf{a}$ is low-dimensional then we can evaluate actions $\mathbf{a}$ on a dense grid. All low-dimensional experiments use a 1D action and we therefore obtain the optimal action via grid search. If the action is high-dimensional (or if the cost function $c(\boldsymbol{\theta},\mathbf{a})$ is expensive to compute) then one can use generic optimization method to identify the optimal action $\mathbf{a}^*$ (e.g., genetic algorithms, or gradient descent if the cost function is differentiable). We term this algorithm NPE-Monte-Carlo (NPE-MC).

For inference step (2) and (3) have to be repeated for every observation $x_o$ and action, resulting in $M \cdot \#x_o \cdot \#actions$ cost evaluations. If evaluating the cost function $c(\boldsymbol{\theta},\mathbf{a})$ is expensive, this comes with considerable computational costs. At the same time, decreasing $M$ or the number of actions to compute the cost for can significantly impact selection quality of the optimal action.

We note that every SBI approach which returns (approximate) posterior samples could be used in order to estimate the expected cost with a Monte-Carlo estimate. This includes approaches like Approximate Bayesian Computation (ABC) (Csilléry et al., 2010) and neural likelihood (Papamakarios et al., 2019) or likelihood-ratio estimation (Hermans et al., 2020; Miller et al., 2022). Here, we chose NPE because it fully amortizes the cost of inference, whereas Neural Likelihood Estimation (NLE) and Neural Ratio Estimation (NRE) require additional MCMC steps and ABC requires re-computing distances for every observation. We additionally report results using ABC and NLE for posterior estimation, referred to as ABC-MC and NLE-MC respectively.

### 3.2  Bayesian Amortized Decision Making (BAM)

The approach outlined above returns the correct action if the neural density estimator indeed captures the true posterior distribution and if the Monte-Carlo estimate provides a sufficiently accurate estimate of the expected costs. However, it may seem overly complicated to infer the full posterior distribution if one is only interested in the expected cost induced by the posterior. In addition, the above method still requires computing the expected cost from Monte-Carlo simulations for *every* observation, which can become prohibitively expensive if the cost function $c(\boldsymbol{\theta},\mathbf{a})$ is expensive to evaluate (e.g., if one has to run a simulator to evaluate $c(\boldsymbol{\theta},\mathbf{a})$, Alsing et al. (2023)). To overcome these two limitations, we introduce a method which (1) does not require estimating the full posterior and (2) directly returns an estimate for the *expected* cost, thereby circumventing the need to recompute the cost for every observation (Fig. 1). We term this method **B**ayesian **A**mortized decision **M**aking (BAM).

---

**Algorithm 1:** Bayesian Amortized decision Making (BAM)

---

**Inputs:** prior $p(\boldsymbol{\theta})$, simulator with implicit likelihood $p(\mathbf{x}|\boldsymbol{\theta})$, cost function $c(\mathbf{x}, \mathbf{a})$, distribution of actions $p(\mathbf{a})$, number of simulations $N$, feedforward NN $f_\phi$ with parameters $\phi$, NN learning rate $\eta$, observations (not i.i.d.) $\mathbf{x}_o^{\{k=1...K\}}$.

**Outputs:** action $\mathbf{a}^*$ with minimal expected cost given an observation.

**Generate dataset:**
sample prior and simulate: $\boldsymbol{\theta}, \mathbf{x} \leftarrow \{\boldsymbol{\theta}_i \sim p(\boldsymbol{\theta}), \mathbf{x}_i \sim p(\mathbf{x}|\boldsymbol{\theta}_i)\}_{i:1...N}$

**Training:**
**while** *not converged* **do**
    $\mathcal{L} = 0$
    **for** $(\boldsymbol{\theta}, \mathbf{x})$ *in* $\{(\boldsymbol{\theta}_i, \mathbf{x}_i\}_{\{i=1...N\}}$ **do**
        $\mathbf{a} \sim p(\mathbf{a})$
        $\mathcal{L} \mathrel{+}= \frac{1}{|N|}(f_\phi(\mathbf{x}, \mathbf{a}) - c(\boldsymbol{\theta}, \mathbf{a}))^2$
    $\phi \leftarrow \phi - \eta \cdot \mathrm{Adam}(\nabla_\phi \mathcal{L})$

**Inference:**
**for** $k \in [1, ..., K]$ **do**
    $\mathbf{a}^* = \min f_\phi(\mathbf{x}_o^{(k)}, \mathbf{a})$ ;           `// find optimal action, e.g., with grid search`

---

Like NPE, BAM is trained on samples from the joint distribution $p(\boldsymbol{\theta}, \mathbf{x})$. Unlike NPE, however, it employs a feedforward neural network that takes data and independently sampled actions $\mathbf{a} \sim p(\mathbf{a})$ as inputs and aims to predict the cost of the associated underlying parameter set $c(\boldsymbol{\theta}, \mathbf{a})$. We train this feedforward neural network with Mean-Squared-Error (MSE) loss and thereby can recover the expected cost $\mathbb{E}_{p(\boldsymbol{\theta}|\mathbf{x}_o)}[c(\boldsymbol{\theta}, \mathbf{a})]$ under the true posterior for every data-action-pair:

**Proposition 1.** *Let $p(\boldsymbol{\theta}, \mathbf{x})$ be the joint distribution over parameters and data and let $p(\mathbf{a})$ be a distribution of actions whose support covers the range of permissible actions. Let $c(\boldsymbol{\theta}, \mathbf{a})$ be a cost function and $f_\phi(\cdot)$ a parameterized function. Then, the loss function $\mathcal{L}(\boldsymbol{\phi}) = \mathbb{E}_{p(\boldsymbol{\theta}, \mathbf{x})p(\mathbf{a})}[(f_\phi(\mathbf{x}, \mathbf{a}) - c(\boldsymbol{\theta}, \mathbf{a}))^2]$ is minimized if and only if, for all $\boldsymbol{\theta} \in supp(p(\mathbf{x}))$ and all $\mathbf{a} \in supp(p(\mathbf{a}))$ we have $f_\phi(\mathbf{x}, \mathbf{a}) = \mathbb{E}_{p(\boldsymbol{\theta}|\mathbf{x})}[c(\boldsymbol{\theta}, \mathbf{a})]$.*

Proof in Appendix Sec. A2. The algorithm is summarized in algorithm 1. We note that it is important for $p(\mathbf{a})$ to cover the full support of $a$ such that all possible action can be observed during training. We further note that, like NPE-MC, BAM requires a final phase to search for the optimal action that minimizes the expected cost. Importantly, BAM requires only a single forward pass for every action and no further evaluations of the cost function, which renders this step computationally cheap.

## 4 Results

### 4.1 Illustrative example

Inspired by the nuclear power plant example by Lacoste–Julien et al. (2011), we illustrate NPE-MC and BAM on a simple example with one-dimensional parameter space and observations, but which induces a bimodal posterior (Fig. 2a,b). To illustrate potential failure modes of NPE-MC, we parameterize the posterior using a neural spline flow with two splines which fails to accurately capture the true, bimodal posterior and assigns too little probability mass to the left mode (Fig. 2b). We then define a cost function $c(\theta, a)$ which assigns zero cost if the action is optimal and whose cost increases monotonically with distance from the true parameter. Crucially, the cost function is asymmetric: for large $\theta$, the cost is high even if the estimation error is small, whereas for low $\theta$, the cost remains low for a larger region of $\theta$ (Fig. 2c). See Appendix Fig. A3 for more parameterizations of the NPE posterior.

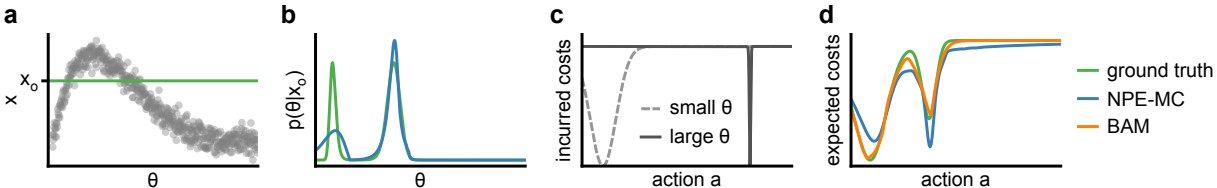

Figure 2: **Illustrative Example.** (a) Joint distribution of parameter and data (gray) and an observation (green). (b) True posterior distribution $p(\boldsymbol{\theta}|\mathbf{x}_o)$ and an estimate obtained with NPE with a NSF with two splines. (c) Cost function for two values of $\boldsymbol{\theta}$ (dotted and full). (d) Expected costs as a function of the action obtained by the ground truth posterior (green) and as estimated with BAM (orange) and NPE-MC (with NSF, blue). Both NPE and BAM were trained with a simulation budget of 5k. See Appendix Fig. A3 for more parameterizations of NPE.

Due to this asymmetry of costs, the approximation to the posterior obtained with NPE systematically underestimates the optimal action, and the optimal action chosen by NPE-MC incurs a much larger cost than the true posterior (Fig. 2d).

## 4.2 Benchmark decisions

Next, we systematically evaluated the performance of BAM, NPE-MC and ABC-MC on tasks where the ground truth posterior is available. We used the toy example introduced above and three previously published simulators with ground truth posteriors (Lueckmann et al., 2021). For all tasks, we defined cost functions on top of which to perform Bayesian decision making (details in Appendix Sec. A4). All cost functions used for the evaluation are based on a flipped bell-shaped curve whose width depends on parameter $\boldsymbol{\theta}$:

$$c(\boldsymbol{\theta}, \mathbf{a}) = 1 - \exp\left(\frac{(\tilde{\boldsymbol{\theta}} - \tilde{\mathbf{a}})^2}{w}\right)$$

where parameters and actions are normalized to take values in the same range, $\tilde{\boldsymbol{\theta}} = (\boldsymbol{\theta} - \boldsymbol{\theta}_{min})/(\boldsymbol{\theta}_{max} - \boldsymbol{\theta}_{min})$, $\tilde{\mathbf{a}} = (\mathbf{a} - \mathbf{a}_{min})/(\mathbf{a}_{max} - \mathbf{a}_{min})$ and the width of the bell curve is determined by $w = (f \cdot \frac{1}{|\boldsymbol{\theta} - o| + \epsilon})^m$.

**Toy example**: A toy simulator with 1D $\boldsymbol{\theta}$ and 1D $\mathbf{x}$ (Fig. 2a). We use the cost function described above with $f = 2$, $m = 2$. We note that, for the benchmark results, we used a neural spline flow as approximate density (Durkan et al., 2019).

**Linear Gaussian**: Simulator with a 10D Gaussian prior and linear Gaussian likelihood, leading to a Gaussian posterior. We use the a similar cost function as for the toy example, but with $f = 0.5$ and $m = 2$ and offset $o = |\boldsymbol{\theta}_{max}|$ to increase sensibility of the cost function towards small and large parameter values.

**SIR**: The SIR (suspectible-infected-removed) model is an epidemiological model that describes the basic spreading behaviour of an infectious disease that is transmitted from human to human over time. The simulator is parameterized by two parameters, the contact rate $\beta$ and the recovery rate $\gamma$. The simulator returns the number of infected individuals $I$ at 10 evenly-spaced points in time. For the decision task we aim to estimate the reproduction rate as the ratio $\mathcal{R}_0 = \frac{\beta}{\gamma}$. We assume that it is particularly crucial to estimate the value of $\mathcal{R}$ around 1 and, therefore, define a cost function which is more sensitive to deviations from the optimal action for $\mathcal{R}_0 \approx 1$ ($f = 2$, $m = 2$, $o = 1$, details in Appendix Sec. A4).

**Lotka-Volterra**: The Lotka-Volterra model describes the oscillating dynamics between two species in a predator-prey relationship. These dynamics are determined based on four parameters: the prey population grows at rate $\alpha$ and shrinks at rate $\beta$. In the presence of prey, predators are born at rate $\delta$ and die at rate $\gamma$. We evaluated four different cost functions with $f = 3$, $m = 2$, where each individual cost function assesses only the accuracy of the estimated marginal distribution (i.e., the cost function ignores three out of four parameters). Analogous to the toy example, the costs function is asymmetric and depends on the underlying true parameter value. The higher the parameter value, the more accurate the prediction of the optimal action

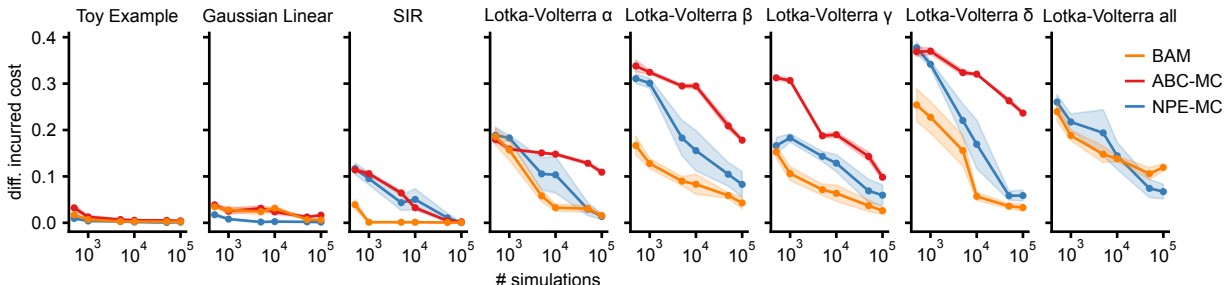

Figure 3: **Benchmarking decision tasks.** Gap between the incurred cost of optimal actions computed from the ground truth posterior and the predicted optimal actions of BAM (orange), NPE-MC (blue) and ABC-MC (red) as a function of the simulation budget. Results are averaged across ten observations and five seeds. Shaded areas show the standard error of the mean across seeds. Columns: four different simulators (with four different decision tasks for Lotka-Volterra).

Table 1: **Runtimes for NPE-MC and BAM.** Exemplary runtimes for the multi-dimensional Lotka-Volterra task measured on a NVIDIA GeForce RTX 2080 Ti GPU. Training dataset size is $N = 10k$, inference runtime is computed based on $10k$ (data, action) pairs. For NPE, $M = 1000$ samples were drawn from the posterior. Mean and standard deviation are based on five seeds.

| method | training | inference per (x, a) pair |
|---|---|---|
| NPE-MC | $1041.0525 \pm 230.2177\,\text{s}$ | $81.4303 \pm 5.1675\,\text{μs}$ |
| BAM | $17.5475 \pm 1.8723\,\text{s}$ | $0.6816 \pm 0.0408\,\text{μs}$ |

has to be to incur small costs. Intuitively, if the action entails various measures to intervene on the rate of shrinkage or growth, this means that overestimation of the rate should be prevented.

**Lotka-Volterra (4d)**: In addition to the low-dimensional tasks described above, we add a multi-dimensional setup based on Lotka-Volterra. Using all four parameters $\theta = \{\alpha, \beta, \gamma, \delta\}$, we also define a four-dimensional action space and define the costs to be the mean incurred costs over all parameters, where individual cost functions are characterized in the same way as described above.

For all tasks, we trained NPE-MC, ABC-MC and BAM with 6 different simulation budgets: 500, 1k, 5k, 10k, 50k, and 100k and averaged results over five random seeds. We then computed the Bayes optimal action for ten different observations with all algorithms as outlined in Sec. 3, computed the incurred cost associated with these actions (Sec. 2.1), and compared this to the incurred cost of the optimal action obtained from the ground truth posterior (Fig. 3). For NPE-MC, we computed the expected costs based on $M = 1000$ samples from the deep neural density estimator (but using $M = 10$ and $M = 100$ impacts the results only weakly, see Appendix Fig. A1). For ABC-MC, we used Rejection ABC with only accepting the top 100 samples for the posterior approximation. For BAM, we resampled one action per $(\boldsymbol{\theta}, \mathbf{x})$ pair at every epoch (keeping the same action at every epoch deteriorates performance, see Appendix Fig. A1). See Appendix Fig. A2 for additional results using NLE-MC in place of to NPE-MC.

We first applied NPE-MC, ABC-MC and BAM to the statistical task of the 1D toy example and the linear Gaussian simulator. On the simple 1D toy task all algorithms perform well and lead to essentially optimal actions after around 1k simulations. Similarly, on the linear Gaussian task, the algorithms perform very well and lead to actions with little associated cost. NPE-MC slightly outperforms BAM on this task. We expect that this is because the neural density estimator employed by NPE-MC and ABC-MC has to learn an affine mapping from data to parameters (because the task is a *linear* Gaussian task), whereas BAM has to learn a non-linear function of the actions (which are used during training of BAM).

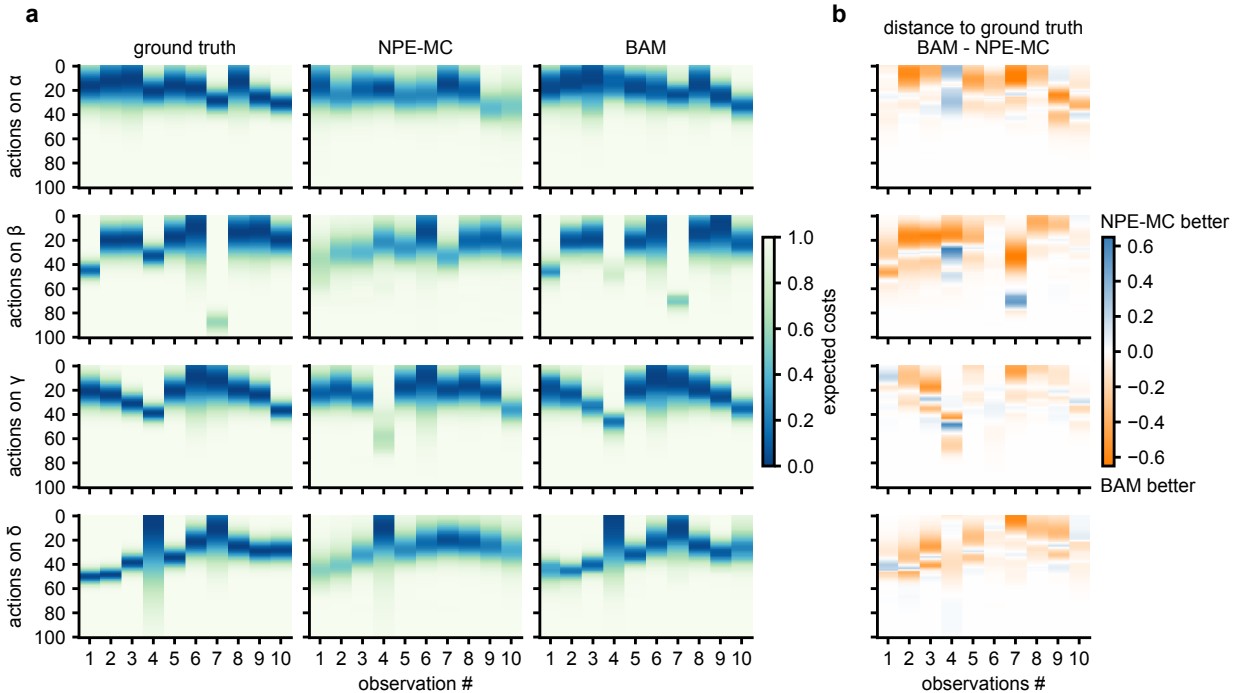

Figure 4: **Detailed comparison of NPE-MC and BAM on Lotka-Volterra task.** (a) Expected costs induced by the ground truth posterior (left), NPE-MC (middle), and BAM (right), for four different cost functions (rows) based on the Lotka-Volterra simulator and a simulation budget of 10k simulations. The x-axis shows the index of the observation, taken from (Lueckmann et al., 2021). (b) Difference in expected costs between NPE-MC and BAM.

We then applied all algorithms to the more complex simulators of the SIR model and the Lotka-Volterra simulator. On both of these simulators and for both algorithms, the induced costs are significantly higher as compared to the previous tasks. For the SIR model, the actions obtained by BAM induce significantly less cost as compared to both NPE-MC and ABC-MC. In order to achieve a similar level of induced cost, NPE-MC often requires one order of magnitude more simulations than BAM. Similarly, for Lotka-Volterra, BAM outperforms both NPE-MC and ABC on all decisions and for all simulation budgets. For fixed simulation budget, the cost induced by BAM can be almost one order of magnitude less than for NPE-MC (e.g. on the SIR task for $10^4$ simulations).

Finally, we applied NPE-MC and BAM to the four-dimensional variant of Lotka-Volterra. For this task, we found that BAM consistently outperforms NPE-MC for lower simulation budgets. For very large simulation budgets, we found NPE-MC to improve over BAM. Runtimes for training and inference show that BAM outperforms NPE-MC in terms of computational costs (Table 1).

In order to gain insights into the strong performance of BAM compared to NPE-MC, we evaluated the expected costs for individual observations and for all possible actions for BAM and NPE-MC for the Lotka-Volterra simulator (Fig. 4). Across all four decision tasks, BAM closely matches the expected costs induced by the ground truth posterior for *all* actions, whereas NPE-MC often fails to capture the expected costs. In particular, for almost every action and observation, BAM provides a closer approximation to the ground truth expected cost than NPE-MC (Fig. 4b). This failure of NPE-MC occurs because NPE does not accurately capture the true posterior, and is not due to the Monte-Carlo estimation procedure (Appendix Fig. A1).

Overall, our results demonstrate that BAM performs similarly to NPE-MC on simple statistical tasks. On more complex and nonlinear simulators, BAM outperformed NPE-MC, sometimes by a large margin. This indicates that, on tasks where estimating the posterior is difficult in itself, it is possible to save a large fraction of simulations if only the Bayes-optimal action is desired (as compared to the full posterior).

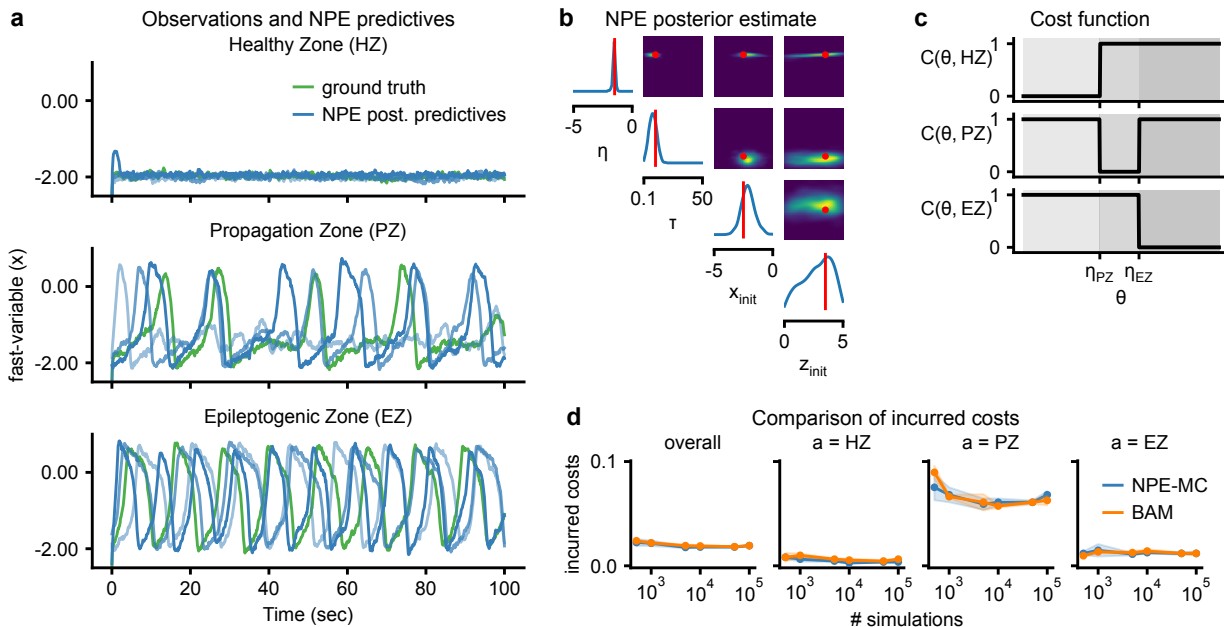

Figure 5: **Decision making for the Bayesian Virtual Epileptic Patient.** (a) Three observations (green, different rows) and posterior predictives obtained with NPE (different shades of blue) for each of the observations. Each row corresponds to one of the zones of the data when classifying by excitability value $\eta$ of the brain region (healthy, propagation, epileptogenic). (b) Marginals and pairwise marginals of the posterior distribution obtained with NPE given an observation from the epileptogenic zone. True parameter values are shown in red. (c) The cost function for predicting $\boldsymbol{\theta}$ to be in one of the three zones (zones shown in gray scale, one decision per row). (d) Incurred costs of NPE-MC and BAM, averaged over 1k prior predictives (left), as well as split by the decision made. Costs are shown for simulation budgets between 500 and 100k simulations. Shaded areas show the standard error of the mean across seeds.

### 4.3 Bayesian Virtual Epileptic Patient

Finally, we performed Bayesian decision making on a real-world task from medical neuroscience. The framework, termed the Bayesian Virtual Epileptic Patient (BVEP), allows to simulate neural activity in a connected network of brain regions to model epilepsy spread (Hashemi et al., 2020; Jirsa et al., 2017). Previous work has demonstrated that NPE is close to an approximate ground truth (obtained with Hamiltonian Monte-Carlo) in this model (Hashemi et al., 2023). The dynamics are controlled by a simulator called 'Epileptor' and in this work, we focus on a single isolated brain region. Based on the level of excitability $\eta$ (one of the free parameters) simulations either lead to simulated data that resembles healthy brains or to activity of brains during an epileptic seizure (Fig. 5a).

The simulator has four free parameters and generates a time series which is reduced into ten summary statistics (details in Appendix Sec. A5, NPE posterior in Fig. 5b). The goal is to predict, given observations, which of three actions is most appropriate (Hashemi et al., 2023). The first action corresponds to no intervention (i.e., a healthy brain region, healthy zone, HZ), the second action corresponds to further tests (unclear if the brain region will be recruited during an epileptic seizure, propagation zone, PZ), and the third action corresponds to an intervention such as surgery (if the brain is clearly unhealthy and triggers seizures autonomously, epileptogenic zone, EZ). We assume that, if the excitability of the brain tissue $\eta$ were known exactly, then one could exactly infer the optimal action and that the optimal action would induce zero 'cost'. For simplicity, we also assume that the cost of a mis-classification (i.e., predict an action which does not match the excitability region of the true parameters) induces a constant cost (Fig 5c). We use this simple cost function to demonstrate that both NPE-MC and BAM can generally solve this task and to avoid making (potentially controversial) assumptions.

We train NPE-MC and BAM on this simulator with simulation budgets ranging from 500 to 100k. For training BAM, we sample actions from a discrete distribution over the three possible actions. After training, we evaluate the expected costs under the three possible actions for both algorithms (we use 1k Monte-Carlo samples for NPE). BAM and NPE-MC both converge with only a few thousand simulations and lead to low expected cost (Fig. 5d, left). We then evaluated both algorithms for cases where the ground truth parameter lies in either of three zones (HZ, PZ, EZ). Both algorithms perform similarly well in each zone, and have the highest cost in cases where the ground truth parameter lies in the propagation zone.

Overall, these results demonstrate that NPE-MC and BAM can both be used to perform Bayesian decision making on the BVEP real-world simulator and that the resulting actions can be obtained with few simulations and induce low cost.

## 5    Discussion

We presented methods to perform amortized Bayesian decision making for simulation-based models. The first method, NPE-MC, uses the parametric posterior returned by neural posterior estimation to infer the optimal action with a Monte-Carlo average. However, this method requires to compute the cost for many parameter sets and actions for every observation, and its performance can suffer if the parametric posterior approximation is poor. Using other methods in place of NPE to estimate the posterior distribution comes with the additional disadvantage that they not fully amortize the cost of inference. To overcome the limitations of NPE-MC, we developed a second method, BAM (**B**ayesian **A**mortized decision **M**aking), which circumvents the need to compute an explicit posterior approximation and directly returns the expected cost. We demonstrated that this method can largely improve accuracy over NPE-MC on challenging benchmark tasks and we demonstrated that both methods can infer decisions for the Bayesian Virtual Epileptic Patient.

**Related Work**    Most applications of Bayesian decision making rely on using Markov-chain Monte Carlo or variational inference methods to obtain an estimate of the posterior distribution. Like NPE, variational inference requires a parametric estimate of the posterior distribution, and Lacoste–Julien et al. (2011) have studied how such an approximation can impact the quality of downstream actions and proposed to take the decision task into account for training the variational posterior. Indeed, if the decision task and the associated loss-function are already known at inference time, then the 'quality' of a posterior approximation can be assessed by the accuracy with which it allows computation of expected costs and optimal actions– some approximation errors will not impact the expected cost, whereas others might have a substantial impact. This makes it possible to quantify the quality of different approximations, and potentially also directly optimize the posterior approximations for this, an approach referred to as 'loss-calibration' (Lacoste–Julien et al., 2011). Several works applied this principle to classification tasks with Bayesian Neural Networks (Cobb et al., 2018), regression tasks for variational inference (Kuśmierczyk et al., 2019) and in the context of importance sampling (Abbasnejad et al., 2015). Instead of adapting the training process of variational inference, which requires to recompute the posterior approximation whenever specifics of the decision task change, Kuśmierczyk et al. (2020) and Vadera et al. (2021) propose post-hoc correction procedures that calibrate the decision making process itself (given an approximate posterior). We expect that, in cases where the parametric posterior approximation of NPE-MC is poor, similar methods could be used to calibrate the NPE posterior for the decision task at hand. BAM, on the other hand, circumvents the need for an explicit posterior approximation and thus does not make any parametric assumptions about the posterior distribution.

Finally, another work on the topic of Bayesian decision making for simulators has been developed concurrently to our work (Alsing et al., 2023). Alsing et al. (2023) focus on simulators which incorporate a mechanistic model for the actions, thereby requiring to simulate in order to estimate the costs. For estimating the cost, our method BAM parallels their method, and they introduce an additional algorithm to estimate the distribution of cost (instead of directly estimating its expected value). Alsing et al. (2023) also introduce a method to focus on inference on a particular observation with sequential inference (Papamakarios and Murray, 2016; Lueckmann et al., 2017; Greenberg et al., 2019; Deistler et al., 2022), whereas we always sample parameters from the prior to avoid correction steps in the loss function. We provide empirical results across eight benchmark decision tasks and for several simulation budgets (Sec. 4.2) and we empirically demonstrate the ability to amortize across observations, whereas Alsing et al. (2023) provide empirical results for single

observations on two benchmark problems. Finally, we provide a proof of convergence for BAM (Appendix Sec. A2), and we apply the methods to Bayesian Virtual Epileptic Patient (Sec. 4.3).

**Limitations**   For any real-world problem, it is difficult to assess the accuracy of the action proposed by either NPE-MC or BAM. In particular, if the real-world observation differs from the simulated data (i.e., if the observation is misspecified), then the neural networks employed by NPE-MC and BAM might give poor results, and any estimated performance on the simulated data might no longer be correct (Cannon et al., 2022; Gao et al., 2023).

In the following paragraphs we briefly outline further considerations regarding the space of actions and the cost function in more detail.

**Action Space**   In Bayesian decision theory the space of actions is assumed to be known to the decision maker at inference time. Unlike NPE-MC, BAM requires the space of available actions to be fixed already at training time as BAM is specifically tailored to the goal of decision making. We thus introduced a distribution of actions $p(a)$ which defines the range of actions across which to amortize, and require the support of $p(a)$ to cover all possible actions. We provided a proof that BAM converges to the expected costs for any action in the support of $p(a)$ and any data in the support of $p(x)$, regardless of the exact form of $p(a)$. Nonetheless, we note that, different distributions $p(a)$ might influence the results in practice. Being amortized over the actions, BAM can can provide insights into the overall cost landscape. This can provide a helpful source of information to guide the final choice of the action, especially if there are additional trade-offs or constraints associated with different actions that are not considered in the simulator or the cost function.

**Cost Function**   The cost function determines which inaccuracies hurt the decision process take the decision task into account, motivating the call for loss-calibration. Another aspect to consider is the computational cost of evaluating the cost function as the proposed methods differ on when to compute the costs of (data, action) pairs: Training of NPE-MC is cost-agnostic, but at inference, the costs have to be computed for every data $x_o$, action $a$ and the number of samples drawn from the posterior estimate $M$. When evaluating the cost function is expensive, e.g., in scenarios where evaluating the cost function requires to simulate (Alsing et al., 2023), this comes with considerable computational costs. In contrast, BAM requires evaluations of the cost function only during training, in total $\#epochs \cdot (N_{train} + N_{val})$ evaluations. At inference, BAM requires no further evaluation of the cost function, but $\#actions$ many passes through the network, which makes this step is computationally very cheap. For high-dimensional actions in particular, the amortization over the decision process makes BAM largely more efficient than NPE-MC.

**Whether to use NPE or BAM**   We presented two approaches to perform amortized Bayesian decision making, NPE-MC and BAM. Their key difference is that NPE-MC builds a full parametric estimate of the posterior distribution (which is independent of the cost function or the specific action), whereas BAM circumvents the requirement to build a posterior approximation and immediately regresses onto the expected cost for every action in the support of $p(a)$ and for every observation in the support of $p(x)$. The key advantages of NPE-MC are that it provides a full posterior that is interpretable and can be used for multiple downstream applications (one of which might be Bayesian decision making), and that it does not require retraining when the cost function or the space of actions changes. Contrary to that, we expect BAM to shine especially in applications where the posterior is complex and hard to estimate (as evidenced by the results on Lotka-Volterra and SIR). In addition, BAM immediately regresses onto the *expected* cost for every action in the support of $p(\mathbf{a})$ and every observation in the support of $p(\mathbf{x})$, whereas NPE-MC requires additional Monte-Carlo simulations and additional evaluations of the cost function for every observation, action, and parameter set.

**Conclusions**   We developed BAM, an algorithm to perform amortized Bayesian decision making. We demonstrated that the approach recovers the Bayes-optimal action for a diverse set of tasks and that it can be used to perform optimal decision making in an example from the medical neurosciences. In addition, we provided a systematic evaluation of commonly used methods from the field of simulation-based inference, namely NPE, ABC and NLE, in the context of decision making on a diverse set of tasks.

## Acknowledgements

MG and MD are supported by the International Max Planck Research School for Intelligent Systems (IMPRS-IS). MD and JHM are funded by the Machine Learning Cluster of Excellence, EXC number 2064/1–390727645. This work was supported by the Tübingen AI Center (Agile Research Funds), the German Federal Ministry of Education and Research (BMBF): Tübingen AI Center, FKZ: 01IS18039A. We thank Julius Vetter, Guy Moss, and Manuel Gloeckler for discussions and comments on the manuscript.

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

# Appendix

## A1 Code and Reproducibility

Code to reproduce all experiments, including the full git commit history, is available at `https://github.com/mackelab/amortized-decision-making`. All neural networks and optimization loops are written in pytorch (Paszke et al., 2019). We tracked all experiments with hydra (Yadan, 2019). For NPE, we used the implementation in the sbi toolbox (Tejero-Cantero et al., 2020).

## A2 Convergence proof for proposition 1

The proof closely follows standard proofs that regression converges to the conditional expectation.

*Proof.* We aim to prove that

$$
\mathbb{E}_{\theta, \mathbf{x} \sim p(\boldsymbol{\theta}, \mathbf{x}), \mathbf{a} \sim p(\mathbf{a})}[(c(\boldsymbol{\theta}, \mathbf{a}) - g(\mathbf{x}, \mathbf{a}))^2] \geq
$$
$$
\mathbb{E}_{\theta, \mathbf{x} \sim p(\boldsymbol{\theta}, \mathbf{x}), \mathbf{a} \sim p(\mathbf{a})}[(c(\boldsymbol{\theta}, \mathbf{a}) - \mathbb{E}_{\boldsymbol{\theta}' \sim p(\boldsymbol{\theta}|\mathbf{x})}[c(\boldsymbol{\theta}', \mathbf{a})])^2]
$$

for every function $g(\mathbf{x}, \mathbf{a})$. We begin by rearranging expectations:

$$
\mathbb{E}_{\theta, \mathbf{x} \sim p(\boldsymbol{\theta}, \mathbf{x}), \mathbf{a} \sim p(\mathbf{a})}[(c(\boldsymbol{\theta}, \mathbf{a}) - g(\mathbf{x}, \mathbf{a}))^2] =
$$
$$
\mathbb{E}_{p(\mathbf{a})}[\mathbb{E}_{\theta, \mathbf{x} \sim p(\boldsymbol{\theta}, \mathbf{x})}[(c(\boldsymbol{\theta}, \mathbf{a}) - g(\mathbf{x}, \mathbf{a}))^2]]
$$

Below, we prove that, for a given $\mathbf{a}$ and for *any* $\mathbf{x}$, the optimal $g(\mathbf{x}, \mathbf{a})$ is the conditional expectation $\mathbb{E}_{\boldsymbol{\theta}' \sim p(\boldsymbol{\theta}|\mathbf{x})}[c(\boldsymbol{\theta}', \mathbf{a})]$:

$$
\mathbb{E}_{\theta, \mathbf{x} \sim p(\boldsymbol{\theta}, \mathbf{x})}[(c(\boldsymbol{\theta}, \mathbf{a}) - g(\mathbf{x}, \mathbf{a}))^2] =
$$
$$
\mathbb{E}_{\theta, \mathbf{x} \sim p(\boldsymbol{\theta}, \mathbf{x})}[(c(\boldsymbol{\theta}, \mathbf{a}) - \mathbb{E}_{\boldsymbol{\theta}' \sim p(\boldsymbol{\theta}|\mathbf{x})}[c(\boldsymbol{\theta}', \mathbf{a})] + \mathbb{E}_{\boldsymbol{\theta}' \sim p(\boldsymbol{\theta}|\mathbf{x})}[c(\boldsymbol{\theta}', \mathbf{a})] - g(\mathbf{x}, \mathbf{a}))^2] =
$$
$$
\mathbb{E}_{\theta, \mathbf{x} \sim p(\boldsymbol{\theta}, \mathbf{x})}[(c(\boldsymbol{\theta}, \mathbf{a}) - \mathbb{E}_{\boldsymbol{\theta}' \sim p(\boldsymbol{\theta}|\mathbf{x})}[c(\boldsymbol{\theta}', \mathbf{a})])^2 + (\mathbb{E}_{\boldsymbol{\theta}' \sim p(\boldsymbol{\theta}|\mathbf{x})}[c(\boldsymbol{\theta}', \mathbf{a})] - g(\mathbf{x}, \mathbf{a}))^2] + X
$$

with

$$
X = \mathbb{E}_{\theta, \mathbf{x} \sim p(\boldsymbol{\theta}, \mathbf{x})}[(c(\boldsymbol{\theta}, \mathbf{a}) - \mathbb{E}_{\boldsymbol{\theta}' \sim p(\boldsymbol{\theta}|\mathbf{x})}[c(\boldsymbol{\theta}', \mathbf{a})])(\mathbb{E}_{\boldsymbol{\theta}' \sim p(\boldsymbol{\theta}|\mathbf{x})}[c(\boldsymbol{\theta}', \mathbf{a})] - g(\mathbf{x}, \mathbf{a}))],
$$

By the law of iterated expectations, one can show that $X = 0$:

$$
X = \mathbb{E}_{\mathbf{x}' \sim p(\mathbf{x})} \mathbb{E}_{\mathbf{x} \sim p(\mathbf{x})} \mathbb{E}_{\boldsymbol{\theta} \sim p(\boldsymbol{\theta}|\mathbf{x})}[(c(\boldsymbol{\theta}, \mathbf{a}) - \mathbb{E}_{\boldsymbol{\theta}' \sim p(\boldsymbol{\theta}|\mathbf{x})}[c(\boldsymbol{\theta}', \mathbf{a})])(\mathbb{E}_{\boldsymbol{\theta}' \sim p(\boldsymbol{\theta}|\mathbf{x})}[c(\boldsymbol{\theta}', \mathbf{a})] - g(\mathbf{x}, \mathbf{a}))].
$$

The first term in the product reads a difference of the same term, and, thus, is zero. Thus, since $(\mathbb{E}_{\boldsymbol{\theta}' \sim p(\boldsymbol{\theta}|\mathbf{x})}[c(\boldsymbol{\theta}', \mathbf{a})] - g(\mathbf{x}, \mathbf{a}))^2 \geq 0$, we have:

$$
\mathbb{E}_{\theta, \mathbf{x} \sim p(\boldsymbol{\theta}, \mathbf{x})}[(c(\boldsymbol{\theta}, \mathbf{a}) - g(\mathbf{x}, \mathbf{a}))^2] \geq \mathbb{E}_{\theta, \mathbf{x} \sim p(\boldsymbol{\theta}, \mathbf{x})}[(c(\boldsymbol{\theta}, \mathbf{a}) - \mathbb{E}_{\boldsymbol{\theta}' \sim p(\boldsymbol{\theta}|\mathbf{x})}[c(\boldsymbol{\theta}', \mathbf{a})])^2]
$$

Because this inequality holds for *any* $\mathbf{a}$, the average over $p(\mathbf{a})$ will also be minimized if and only if $g(\mathbf{x}, \mathbf{a})$ matches the conditional expectation $\mathbb{E}_{\boldsymbol{\theta}' \sim p(\boldsymbol{\theta}|\mathbf{x})}[c(\boldsymbol{\theta}', \mathbf{a})]$ for any $\mathbf{a}$ within the support of $p(\mathbf{a})$. □

## A3 Experimental setup

We used mainly the same hyperparameters for all benchmark tasks and for the Bayesian Virtual Epileptic Patient task. Only the learning rate was adjusted to 0.005 for Lotka-Volterra, while it was set to 0.001 for all other tasks.

For NPE, we used the default parameters of the sbi package (Tejero-Cantero et al., 2020), apart from the density estimator, for which we used a neural spline (Durkan et al., 2019). Following Hashemi et al. (2023), we used a masked autoregressive flow (Papamakarios et al., 2017) for the Bayesian Virtual Epileptic Patient with default parameters as set in the sbi package.

For BAM, we used a feedforward residual network (He et al., 2016) with 3 hidden layers and 50 hidden units each. We use ReLU activation functions and, for cases where we know that the expected cost is going to be positive and bounded by 1, squash the output through a sigmoid. We use the Adam optimizer (Kingma and Ba, 2015). As described in Sec. 4, the size of the training dataset varied between 500 and 100k. In all cases, the training dataset was split 90:10 into training and validation and with a batchsize of 500.

As distribution over all permissible actions, $p(\mathbf{a})$, we use a uniform distribution over continuous actions between 0 and 100 for all benchmark tasks, $\mathcal{U}(0, 100)$, and a discrete uniform distribution over three classes for the Bayesian Virtual Epileptic Patient.

## A4 Benchmark tasks

Below, we provide details on the simulators and cost functions used for benchmarking NPE-MC and BAM.

**Toy example**: The prior follows a uniform distribution

$$p(\boldsymbol{\theta}) = \mathcal{U}[0, 5]$$

and the likelihood is given by:

$$p(\mathbf{x}|\boldsymbol{\theta}) = 50 + 0.5\boldsymbol{\theta}(5 - \boldsymbol{\theta})^4 + \nu,$$

where $\nu \sim \mathcal{N}(\theta, 10)$. We obtained (an approximation to) the ground truth posterior via quadrature.

As cost function, we used a flipped (along the y-axis) bell-shaped curve whose width decreases with increasing parameter $\boldsymbol{\theta}$:

$$c(\boldsymbol{\theta}, \mathbf{a}) = 1 - \exp\left(\frac{(\boldsymbol{\theta} - \mathbf{a})^2}{(\frac{2}{|\boldsymbol{\theta}|+\epsilon})^2}\right)$$

with $\epsilon = 0.1$. As the parameter space and the action space differ, $\boldsymbol{\theta}$ and $\mathbf{a}$ were rescaled to both lie within the range of $[0, 10]$.

**Linear Gaussian**: We used exactly the same simulator as in Lueckmann et al. (2021). As cost function, we used a flipped (along the y-axis) bell-shaped curve whose width decreases towards the extremes of the parameter range:

$$c(\boldsymbol{\theta}, \mathbf{a}) = 1 - \exp\left(\frac{(\boldsymbol{\theta} - \mathbf{a})^2}{(\frac{0.5}{|\boldsymbol{\theta}-0.5\cdot(\boldsymbol{\theta}_{max}-\boldsymbol{\theta}_{min})|+\epsilon})^2}\right)$$

with $\epsilon = 0.1$. As the parameter space and the action space differ, $\boldsymbol{\theta}$ and $\mathbf{a}$ were rescaled to both lie within the range of $[0, 10]$. Additionally, a shift was introduced such that the variance is widest at 5 and decreases towards both ends of the value range.

**Lotka-Volterra:** We used exactly the same simulator as in Lueckmann et al. (2021). We evaluated four different cost functions, where each individual cost function assesses only the accuracy of the estimated marginal distribution (i.e., the cost function ignores three out of four parameters). The cost function for individual parameters $\theta_i$ with $i = 0, 1, 2, 3$ is given by:

$$c(\boldsymbol{\theta}_i, \mathbf{a}) = 1 - \exp\left(\frac{(\boldsymbol{\theta}_i - \mathbf{a})^2}{(\frac{3}{|\boldsymbol{\theta}|+\epsilon})^2}\right)$$

with $\epsilon = 0.1$. As the parameter space and the action space differ, $\boldsymbol{\theta}$ and $\mathbf{a}$ were rescaled to both lie within the range of $[0, 10]$.

**SIR:** We used exactly the same simulator as in Lueckmann et al. (2021). The cost function is based on the ration between parameters, $\mathbf{r} = \frac{\beta}{\gamma}$, and is given by:

$$c(\mathbf{r}, \mathbf{a}) = 1 - \exp\left(\frac{(\mathbf{r} - \mathbf{a})^2}{(\frac{2}{|10-|\mathbf{r}-1||+\epsilon})^2}\right)$$

with $\epsilon = 0.1$. As the parameter space and the action space differ, $\mathbf{r}$ and $\mathbf{a}$ were rescaled to both lie within the range of $[0, 10]$. Additionally a shift by 1 in parameter space was introduced such that the cost function is very sensitive around $\mathbf{r} = 1$ and variance increases with increasing distance to 1.

## A5 Details on Bayesian Virtual Epileptic Patient (BVEP) simulator

The Bayesian Virtual Epileptic Patient provides a framework to simulate neural activity in a connected network of brain regions to model epilepsy spread (Hashemi et al., 2020; Jirsa et al., 2017). The dynamics of individual brain regions are controlled by a simulator called 'Epileptor' that allows to reproduce dynamics of electrical activity during seizure-like events. Hashemi et al. (2020) introduce two versions of the Epileptor, the full 6D model and a reduced variant, the 2D Epileptor. The full Epileptor model (Jirsa et al., 2014; Hashemi et al., 2020) is composed of five state variables that couple two oscillatory dynamical systems on three different time scales. Variables $x_1$ and $y_1$ are associated with fast electrical discharges during seizures at the fastest time scale. Variables $x_2$ and $y_2$ act on an intermediate timescale and the permittivity state variable $z$ that governs the transition between ictal, i.e. during a seizure, and interictal, i.e. in between seizures, states on a slow time scale. See Jirsa et al. (2014) and Hashemi et al. (2020) for a detailed description of the Epileptor equations. For this work, we focus on a single isolated brain region and used the reduced '2D Epileptor' model which results from applying averaging methods to make the effects of $x_2$ and $y_2$ negligible and further assuming time scale separation so that $x_1$ and $y_1$ collapse on the slow manifold (Hashemi et al., 2020).

The 2D Epileptor takes four parameters to generate a time series of neural activity: $\eta$ describes the excitability of the tissue, $\tau$ controls the time scale separation and $x_{init}$ and $z_{init}$ are initial values for the state variables. See Hashemi et al. (2020) for a full description. Based on the level of excitability $\eta$, brain regions can be categorized into three types (Hashemi et al., 2020):

- Epileptogenic Zone (EZ): if $\eta > \eta_{EZ}$, the Epileptor can trigger seizures autonomously

- Propagation Zone (PZ): if $\eta_{PZ} = \eta_{EZ} - \Delta\eta < \eta < \eta_{EZ}$, the Epileptor cannot trigger seizures autonomously, but (in a connected network of brain regions) can be recruited during the evolution of an epileptic seizure

- Healthy Zone (HZ): if $\eta < \eta_{PZ}$, the Epileptor does not trigger seizures

Following Hashemi et al. (2023), we set the critical value $\eta_{EZ} = -2.05$ and $\Delta\eta = 1.0$, i.e. $\eta_{PZ} = \eta_{EZ} - \Delta\eta = -3.05$. These types of brain regions form the basis for the decision task described in Sec. 4.3.

When observations are high-dimensional as it is the case for the time series generated by the Epileptor, low-dimensional summary statistics are commonly used for inference. For both, NPE-MC and BAM, we used 10 time-independent summary statistics to describe the observed neural activity. These are the mean, median, standard deviation, skew, and kurtosis of the time series as well as higher moments (up to the 10th moment of the observation), the power envelope, seizure onset, the amplitude phase, and spectral power.

## A6 Supplementary figures

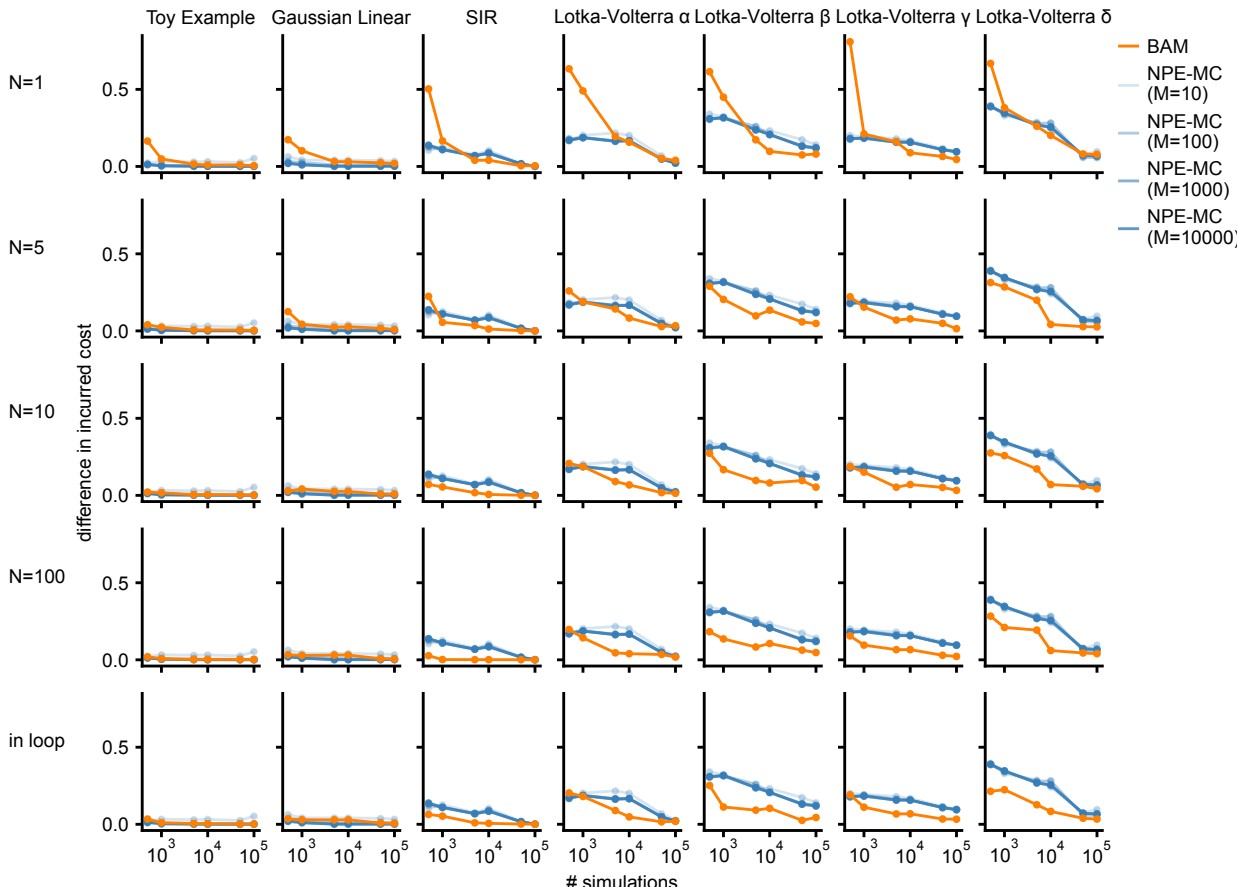

Figure A1: **Expected costs for sampling diverse numbers of actions per data point and for a different number of Monte-Carlo (MC) samples.** NPE-MC is the same for every row (blue). Different shades of blue indicate NPE-MC with different number of MC-samples. For BAM, we sampled $N = \{1, 5, 10, 100\}$ actions per $(\boldsymbol{\theta}, \mathbf{x})$ pair in the training dataset (rows). In the last row we sampled one new action at every epoch. This is shown in the main paper in Fig. 3.

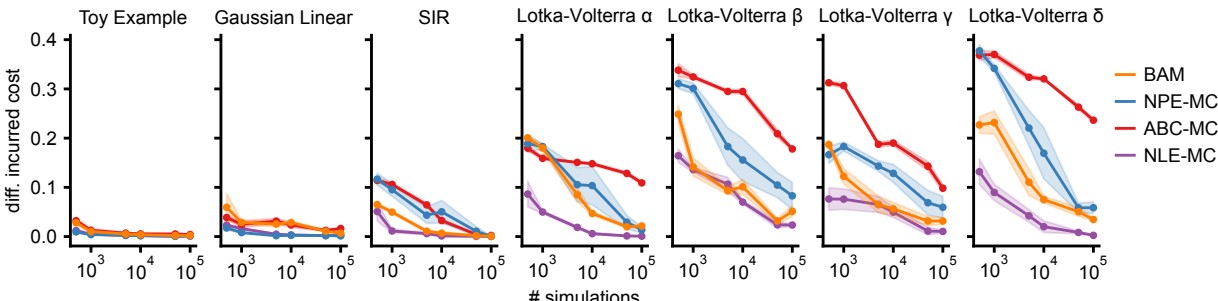

Figure A2: **Benchmarking decision tasks.** Gap between the incurred cost of optimal actions computed from the ground truth posterior and the predicted optimal actions of BAM (orange), NPE-MC (blue), ABC-MC (red) and NLE-MC (purple), averaged across ten observations and five seeds. Columns: four different simulators (with four different decision tasks for Lotka-Volterra). NLE-MC performed well across all benchmark tasks. In particular, we found that NLE-MC outperformed NPE-MC on all benchmark tasks and also BAM for most simulation budgets. We note that it has previously been reported that NLE performs particularly well on these tasks (Lueckmann et al. 2021). These results indicate that, if the cost of inference (e.g., with MCMC) and of estimating the cost (via simulation) is not critical to the user, then NLE-MC can provide a strong alternative.

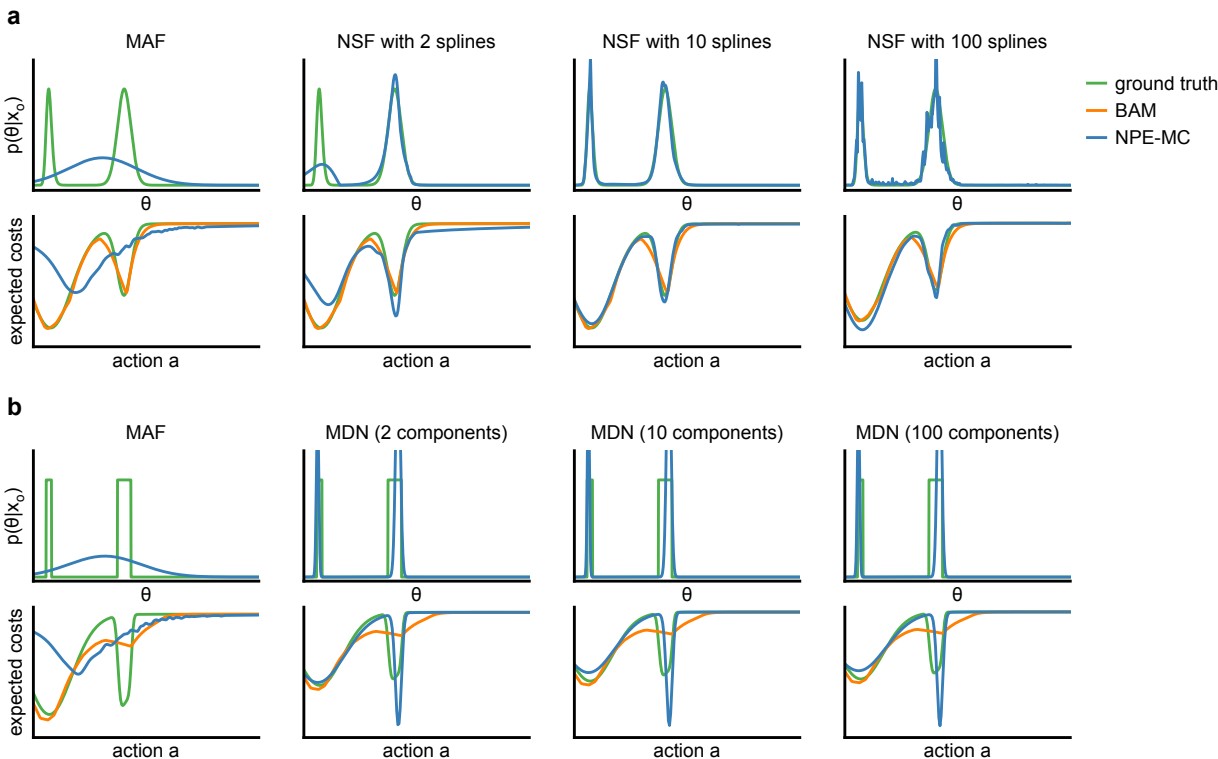

Figure A3: **Illustrative Example.** (a) Simulator with Gaussian noise and (b) Simulator with uniform noise. First row: True posterior distribution $p(\boldsymbol{\theta}|\mathbf{x}_o)$ and an estimate obtained with NPE. Second row: Expected costs as a function of the action obtained by the ground truth posterior (green) and as estimated with BAM (orange) and NPE-MC (blue). Columns correspond to different parametrizations of the posterior estimate. Both NPE-MC and BAM were trained with a simulation budget of 5k.

