# OpenReview forum: "Amortized Bayesian Decision Making for simulation-based models"
_TMLR — Accepted by TMLR_

### Review · Reviewer_NspT · 2024-01-19

**Summary Of Contributions:**

The paper proposes a novel way to do decision-making for simulation-based models. The proposed method, called Bayesian Amortized decision Making (BAM), is based on the idea of amortizing the inference of the action without explicitly approximating the posterior distribution over the parameters of the model. The main motivation is that the posterior over the parameters is often multimodal, and therefore it is difficult to approximate it well. Instead, the authors propose to learn a mapping from the observations and actions, and then use this mapping to select the best action. The mapping is learned using a neural network, and the parameters of the model are sampled from a prior distribution. The proposed method is then compared with a baseline method on a set of experiments (from toy examples to classic benchmarks), showing that it is able to perform well in all the considered scenarios.

**Audience:**

Yes

**Claims And Evidence:**

Yes

**Requested Changes:**

- Armonize "Introduction" and "Background" sections. There is some duplicated content between the two sections, and it would be best if you could find a way to introduce the problem and the background content in a coherent way
- Also related to the point above, please be a bit more rigorous in the mathematical definitions. What is $\boldsymbol x_o$? $\boldsymbol \theta$? Likelihood $p(\boldsymbol x_o|\boldsymbol \theta)$ and prior $p(\boldsymbol \theta)$ are never explicitely introducted.
- Position of Algorithm and Figures closer to the discussion or after most of the content is discussed in the main text (e.g. Fig 1 is in page 2 and discussed in page 5)
- The message of Figure 1 is unclear to me. It seems that the entire diagram is your _unique_ contribution, while actually it shows two different methods. What about using two different colors to indentify the part of the figure related to BAM and the one related to NPE-MC?
- One key element of the proposed algorithm that it's never mentioned in the text (unless I'm wrong) is the prior on the actions $p(\boldsymbol a)$ which only appears in the Algorithm. To me, this is critical and it requires a careful analysis, or at least a discussion on the effect it has on the solution.
- Regarding the experimental setup, I think it would be best if you could be explicit on the cost functions used (since you already have them in the Appendix---a table might be enough). To be such a critical part of the evaluation, the current descriptions are a bit too general.
- Regarding the experiments, are they averaged across multiple seeds? Is it robust to different initializations?

**Minor**

- "NLE" is not defined in the text
- The "Conclusions" section is very short and it does not add much to the discussion. I would suggest to remove it and move the content to the "Discussion" section.
- At the beginning of the "Introduction" section, when you say that "The posterior distribution is intractable for all but the simplest models, and requires methods such as Markov-Chain Monte Carlo (MCMC) to approximate." which it reads as if only MCMC can be used to approximate the posterior. I would suggest to rephrase it to include also variational inference methods (which are mentioned anyways in later sections the paper).


**Open questions**
- I wonder if BAM can be seen as NPE on a different _joint_ model $\{\boldsymbol \theta,\boldsymbol a\}$, with a factorized prior $p(\boldsymbol \theta,\boldsymbol a)=p(\boldsymbol \theta)p(\boldsymbol a)$ and likelihood $p(\boldsymbol x_o|\boldsymbol \theta,\boldsymbol a)$. Can you maybe have a quick comment on this?

**Strengths And Weaknesses:**

**Strengths**:
- Interesting paper that proposes a novel way to do decision-making for simulation-based models. The idea of amortizing the inference of the action without explicitly approximating the posterior  distribution over the parameters of the model is interesting and novel (to the best of my knowledge, without considering the concurrent work highlighted in the paper).
- The proposed method is simple and elegant, and it seems to work well in the experiments.


**Weaknesses**:
- The writing can be improved, some content is duplicated and it lacks some rigor in the mathematical definitions.
- Contributions are unclear (or better, confusing): from reading the abstract, I would say that the contribution of the paper is _only_ BAM, but Section 3 (and Figure 1) puts forward also NPE-MC as contribution ("Our first method is a direct extension"). Personally, I would consider NPE-MC more of a baseline and I would clearly present it as such.
- The experimental evaluation is a bit weak. The authors compare BAM with NPE-MC, but they do not compare with other methods. I do understand the motivation for focusing on NPE-MC, but I think it would be nice to have a comparison with at least one of the other methods mentioned (e.g. an ABC method).

---

### Review · Reviewer_z9cw · 2024-02-23

**Summary Of Contributions:**

Rooted in a simulation-based inference (SBI) setting, this paper presents two methods to perform Bayesian decision making when the model relating parameters $\theta$ to observations $x$ is a black-box, maybe stochastic, simulator. The goal is to find the bayes optimal action $a^{\star}(x)$ that minimise the expected cost of this action given an observation $x$, under the assumed model $p(\theta \mid x)$ implicitly defined by a prior over parameters,
$$a^{\star}(x) = \arg\min_{a} \mathbb{E}_{p(\theta \mid x)}[c(\theta, a)].$$
This decision strategy should enforce the expected cost over the marginal distribution of possible observation is minimal under correct modelling assumption, that is the simulator and prior represent exactly the true generative process.

Authors suggest two methods to solve this problem. The first method, named NPE-MC, trains a neural posterior estimator of $p(\theta \mid x)$, then, given an observation, it approximates the expected cost with Monte Carlo and use grid search to determine an estimate of the optimal action. The second method, named BAM, observes that the estimation of $p(\theta \mid x)$ can often be strictly more complicated than directly estimating the expected cost  of an action given an observation. Thus, BAM directly trains an estimator $f(a, x)$ that maps a pair of observation and action to the associated expected cost. After training, $f(a, x) \approx \mathbb{E}_{p(\theta \mid x)}[c(\theta, a)]$, and obtaining the optimal action is performed by a grid search on the estimator of the expected cost of performing action $a$ given the observation $x$. Clearly, the two methods are consistent in the sense that given perfect training (and expressive enough class of surrogates) and sufficiently fine grid search, they return the Bayes optimal action.

Authors compare the two methods' performance over 4 synthetic tasks derived from standard simulation-based inference benchmarks and decision making problem on virtual epileptic patients. All tasks have low-dimensional action space.
Experiments demonstrate that none of the two methods is strictly superior than the other on all the benchmark. Nevertheless, BAM is overall slightly less data-hungry than MC-NPE on the SBI-inspired benchmarks. Authors conclude BAM is an effective way to perform Bayesian decision making under implicit model (forward black box simulators).

**Audience:**

Yes

**Broader Impact Concerns:**

I think applying this methodology to make eal-world decision would be dangerous as the method could miserably fail under model misspecification. This should probably be discussed.

**Claims And Evidence:**

Yes

**Requested Changes:**

I would suggest the following things to improve the manuscript and convince me this a work of interest to the TMLR audience:
1. Improve intro, motivate with real-world problems.
2. Discuss how and when NPE fails, say why it can be difficult to address, and how BAM can better perform in such settings maybe?
An even better point would be to show that NPE can fail silently (marginal calibration ok, C2ST ok, etc...) under some metrics but give very bad decision process?!
3. Discuss.
4. Tone down your contribution and be more clear about why doing it in a higher dimension-space would add new challenges. Maybe, also explain why there are already important decision problems that are hard to solve even though the action space is low-dimensional. This goes back to remark 1.
5. Please chance that figure, or do not cite NPE-MC there. This would be a very wrong signal send to people not familiar with NPE.
6. Improve the discussion.
7. This one is bit harsh, but I think improving the intro and trying to provide better insights on why applying NPE-MC may be hard could be nice? I simply believe having a real-world experiment where you can show you achieve non-obvious performance would be much stronger. I also understand this is difficult to show... But at the same time, if you cannot show that, I am not sure the paper has any interest to anyone. Indeed, I did not learn much when reading your paper and so I am skeptical another researcher would really gain anything by building on your paper to develop a method that apply to real-world scenario.
8. Please discuss some variance of your estimators and how this can also guide a practitioner to trust or not the decision process.
9. Really unclear how this can really be helpful in practice. In contrast to standard SBI, which is either applied on real-world data that are very well modelled as in some scientific fields, or just on simulation to understand better the modelling assumptions made by the forward model, here I do not totally see these applications. Again, it would be awesome to see a real-world experiment and I would directly remove my concerns.

**Strengths And Weaknesses:**

# Strength
- The paper is well-written, easy to follow, I would even say pleasant to read.
- I believe the paper may be of interest to a community of non-experts in SBI, who encounter such decision making and have a way to simulate the observation process and cost associated with decisions.
- I have no major concerns on the experiments presented in the paper and find they support appropriately the paper's message.
- I also like the argument that, sometimes, maybe it is reasonable to say often, learning the map from state and action directly to expected cost is an easier ML task than learning the posterior instead. It is even more salient if the cost function is expensive to evaluate (the question is, when is it?)

# Weaknesses
Although I enjoy reading the paper, I have several concerns regarding the relevance of publishing the paper in its current form. I am now expressing my main concerns and will suggest some potential modifications to address these in the next box.
1. I think the introduction does not a very good job at motivating why using the first equation to make a choice is reasonable (and applicable) in real-world scenario.
2. They are a bunch of vague statements made about the "inaccuracy" of NPE. Although I agree in principle with these statements, they should be clarified to explain what kind of inaccuracy may hurt the decision process (e.g., over vs under confidence) and when such inaccuracies are hard to avoid.
3. The loss of interpretability behind deciding with BAM vs deciding with NPE-MC should be discussed. It is unclear to me also, why, if the problem targeted has high stakes we couldn't simply train NPE with more simulations. If I were to use such a strategy to make decision I would rather chose a more costly tools which provides me with some interpretability than the black box BAM.
4. It seems the method would not scale well to higher decision space for two reasons: 1. p(a) in algo 1 might be an important design choice; 2. the grid search might be ineffective, or even dangerous, if the decision+observation spaces lead to a complicated relationship between the cost and the pair action/observation.
5. Figure 2 looks like a caricature and seems dis-honest to NPE-MC in my opinion. It is unlikely anyone having some tiny experience in SBI would obtain such a green NPE. It is thus unclear in which extreme scenario BAM is a significantly wiser choice than NPE-MC
6. The claim about how to solve the problem if it is high dimensional are vert hand-wavy. I do not think it is correct to say that using a generic optimisation method to search the space would be a good choice, especially given the high chance to end up in a place where the surrogate models are innacurate.
7. I am not sure how relevant is this paper for any person already aware of SBI as, in the one hand, NPE-MC is a straightforward extension and BAM is just fitting a deterministic surrogate to simulated data.
8. I would expect to see error bars on figure 3 and 5d.
9. It is unclear how this problem setup applies to any real-world problem as it is unlikely modelling assumptions are as correct in the real-world as they are in simulations. The lack of real-world experiments is what causes me this worry.

---

### Review · Reviewer_s4Bh · 2024-03-06

**Summary Of Contributions:**

This work considers the problem of Bayesian Decision Making, i.e. selecting an action with the minimum expected cost, and proposes a solution that avoids explicitly approximating a posterior by fitting a neural network that predicts the expected cost of an action. In situations where the end goal is solely to find the minimal expected-cost action, the proposed method, named BAM, is computationally faster at test time since the neural network avoids having to sample the posterior to approximate the expected cost. It is instead trained to output the expected cost of an action in a single forward pass. Empirical results demonstrate that when the posterior is misspecified, an existing method’s performance in the Bayesian Decision Making task significantly deteriorates, while the same deteriorating is not observed in BAM. When a posterior is correctly specified, the performance of BAM is comparable to existing methods.

**Audience:**

Yes

**Claims And Evidence:**

Yes

**Requested Changes:**

**Critical**

- The toy example presented in figure 2 articulates that severe misspecification is very detrimental to NPE but not to BAM. The toy example is great for gaining intuition, but as it stands now, this is an extreme example. Does more subtle misspecification, perhaps for this example skewness or heavy-tails, still lead to the same degradation in performance? I believe that similar figures to figure 2 in the appendix could be useful. Does the posterior need to be extremely misspecified for NPE to fail while BAM is not affected?  This is an important question for someone considering using BAM since BAM forces the decision-making process to be (and stay) risk-neutral. If only a small performance degradation in NPE is observed, it may be desirable to accept the (unknown) drop in expected performance to maintain having access to an approximate posterior and tailor the decision-making to a different risk profile.

- What happens to the performance/training time of BAM when the cost function is expensive to evaluate? What if the cost function must be approximated? The contribution of BAM is clear in the situations currently simulated, but if cost is more expensive to evaluate then it’s not clear if BAM offers performance benefits. Empirical results for a scenario like this would be desired.

**Not Critical**

- At the end of Section 2 “…it can be applied to any observation without running further simulations and without retraining…”. To me this sentence could be interpreted as suggesting that the neural network can generalize to data not necessarily in the training set. I suggest revising the sentence if this is not implied.

- Adding in the explicit run times of NPE vs BAM would be helpful.

**Strengths And Weaknesses:**

**Strengths**

- The paper is well organized and written. Introduction to subject matter is clear and concise. Contributions are presented in a logical and intuitive order. Saving a broader discussion of related works until section 5 makes the paper accessible to non-experts of the subject matter.
- I find the empirical results to support the main claims of the paper. Below I mention some weaknesses in the set up of the experiments, but I want to emphasize that the main claims are supported. Specifically, when the posterior is difficult to estimate or if it is misspecified, BAM can output actions with noticeably lower cost than NPE.
- To the best of my knowledge, the proof of proposition 1 is correct.

**Weaknesses**

- Actual run times are not reported. It would be helpful to know how the run times of NPE vs BAM, training and inference, compare. Each iteration in training BAM involves evaluating the cost, and so an iteration should be more expensive. If one of the benefits is amortizing the computation, it’s necessary to understand this effect.
- The experiments only consider scenarios where the cost function is computationally inexpensive to evaluate. This is especially beneficial for BAM, which relies on calling this cost function repeatedly during training, while existing methods do not. When the cost function isn’t cheap to evaluate or must be approximated, it’s not clear that BAM does not break down (i.e. is training prohibitively long). This is a concern since one of the main benefits of BAM is to avoid evaluating the cost function repeatedly for many samples of the posterior at inference time. Note that even if BAM breaks down in this setting, I still maintain that there is a useful contribution from this paper. However, this is critical information to report to the community alongside being able to use BAM in the settings already considered.

---

### Author Response · Authors · 2024-03-21
**Response to all reviewers**

We thank all reviewers for taking the time to engage with our work and for providing useful and insightful feedback. We are happy to hear that the reviewers believe that our work is “*interesting and novel*” (NspT), “*well organized and written*” (s4Bh, z9cw), and that our proposed method is “*simple and elegant*” (NspT). We thank the reviewers for this highly positive feedback.

In response to your feedback, we have revised the paper where suggested (with changes tracked in blue) and added several new results to highlight the utility of our work. Among others, we have added additional algorithms to the benchmark tasks and we have added a new task with a multi-dimensional action (see details and additional results in individual responses). We hope that this will further increase the reviewers’ appreciation of the relevance and novelty of our work, and allow them to recommend our manuscript for acceptance.

---

### Decision · Action_Editor_DnyB · 2024-04-23

**Recommendation:** Accept with minor revision

**Comment:**

All three reviewers are in agreement that this paper meets TMLR's acceptance criteria. The review process and converged in a manuscript where all reviewers are aligned in vetting the conclusions. I kindly ask the authors to fill out the runtimes as delineated in the manuscript whenever convenient.

**Audience:**

This paper develops an idea in the space of Bayesian decision making—which overlaps with TMLR's audience.

**Claims And Evidence:**

The reviewers are satisfied with how this manuscripts contributions are presented and the evidence provided to explain their effectiveness. There is an open item around runtimes, which I am asking the authors to fill out with this last revision.